# A Systematic Map of the Research on Disease Modelling for Agricultural Crops Worldwide

**DOI:** 10.3390/plants11060724

**Published:** 2022-03-09

**Authors:** Giorgia Fedele, Chiara Brischetto, Vittorio Rossi, Elisa Gonzalez-Dominguez

**Affiliations:** 1Department of Sustainable Crop Production (DI.PRO.VES.), Università Cattolica del Sacro Cuore di Piacenza, Via Emilia Parmense 84, 29122 Piacenza, Italy; giorgia.fedele@unicatt.it (G.F.); chiara.brischetto@unicatt.it (C.B.); 2Horta srl, Via Gorra 55, 29122 Piacenza, Italy; e.gonzalez@horta-srl.com

**Keywords:** integrated pest management, systematic review, crop protection, food security

## Abstract

In this work, we developed a systematic map to identify and catalogue the literature pertaining to disease modelling for agricultural crops worldwide. Searches were performed in 2021 in the Web of Science and Scopus for papers reporting any type of disease model for 103 crops. In total, 768 papers were retrieved, and their descriptive metadata were extracted. The number of papers found increased from the mid-1900s to 2020, and most of the studies were from North America and Europe. More disease models were retrieved for wheat, potatoes, grapes, and apples than for other crops; the number of papers was more affected by the crop’s economic value than by its cultivated area. The systematic map revealed an underrepresentation of disease models for maize and rice, which is not justified by either the crop economic value or by disease impact. Most of the models were developed to understand the pathosystem, and fewer were developed for tactical disease management, strategic planning, or scenario analysis. The systematic map highlights a variety of knowledge gaps and suggests questions that warrant further research.

## 1. Background

If not managed, diseases of agricultural crops cause substantial economic losses and reduce food security at household, national, and global levels [1]. New plant diseases or adapted plant pathogen genotypes emerge continuously (or re-emerge) and spread worldwide due to global change, climate change, and the large-scale and intensive production of genetically uniform crops [1,2].

With an increasing world population, a major challenge is how to increase food production [3]. Crop pathogens must be controlled to minimize yield losses while keeping pesticide applications at levels that are economically and ecologically sustainable, thus minimizing the risks for humans and the environment. This has motivated the development of integrated pest management (IPM) throughout the world [4]. The core of IPM is the process of decision-making, including the decision as to whether and when plant protection actions are required and/or economically justified; decision making in IPM increasingly depends on the development of disease models, decision rules, and risk algorithms [4,5,6].

In the context of crop protection, a plant disease model is a simplified representation of the relationships between pathogens, crops, and environment (the pathosystem) that cause the development of epidemics; these relationships involve a large number of interactions at different levels of hierarchy over time and/or space [7,8]. Models combining plant, disease, and environmental factors have been developed since the middle 1900s. The early plant disease models, which were developed following an empirical approach [9], consisted of simple rules, graphs, or tables showing relationships between components of the disease cycles (e.g., infection and sporulation) and the concomitant weather conditions [10,11]. Dynamic modelling of plant disease epidemics was introduced in the early 1960s with the conceptualization of the temporal progress of diseases by Van der Plank [12,13] and with the further development of those concepts by Zadoks [14]. After these “milestones”, many plant disease models have been developed by plant pathologists, mathematicians, or statisticians [15]. These models can be characterized based on the modelling approach (empirical vs. mechanistic), whether models combine information on the host, pathogen, environment, or disease, and the components of the disease cycle they account for [15,16,17,18,19]. Unlike mechanistic models, which predict pathogen and disease levels based on underlying processes, empirical models organizing data and standardizing their relationship in terms of mathematical or statistical representation [4].

Unfortunately, few plant disease models have been applied for practical disease control [15,16,20]. Rossi et al. [4] recently considered the reasons for the poor take-up of models in practical crop protection and highlighted the inadequate transparency in mathematical structure of the better part of the models and the need for extensive model validation. In the last decade, the imbalance between the number of models developed and the number applied has decreased. The increase in model application is closely linked with the advances in weather monitoring, data transmission, and automatic data processing. 

In the current study, we developed a systematic map of the current status of the research on plant disease models worldwide, as deduced from the published literature. Systematic mapping is a recently developed method [21,22] for exploring the literature of a broad topic area; it enables the development of a comprehensive database of literature that includes both qualitative and quantitative information. By developing and using a systematic map, we assess the models currently available for different crops and pathogens, and identify their main characteristics. We also identify future needs for the development, validation, and implementation of plant disease models.

## 2. Objective of the Systematic Map

The main objective of our study was to identify and to systematically map scientific literature pertaining to disease modelling for agricultural crops worldwide. We focused on the literature regarding disease prediction models for any crop worldwide.

The database resulting from the literature search has four main uses. First, it can be used to determine how the research on plant disease models has varied over the years and among countries, and to identify the most studied crop systems and disease-causing organisms. Second, the database provides a catalogue of models for researchers and stakeholders; this catalog can be used as a resource for the development of new plant disease models, or as a guide for policymakers to decide where resources should be allocated. Third, the database provides a foundation for future systematic reviews and meta-analyses. Fourth, and perhaps most importantly, the database can be used to identify knowledge gaps and to thereby inform future research.

## 3. Methods

The methods used in the development of our systematic map were adapted from existing systematic map reports [21,22,23]. The process for the systematic map development is summarized in Figure 1.

### 3.1. Search for Relevant Papers

In 2021, we conducted a search of the two most relevant online bibliographic databases: (i) Web of Science Core Collection (http://webofknowledge.com/WOS, (accessed on 26 August 2021)) and (ii) Scopus (https://www.scopus.com/ (accessed on 26 August 2021)). Database searches were conducted in English, however the search was not restricted to papers written in English (i.e., papers with the title and abstract in English, yet with the main document in another language were included).

Search terms relevant for the systematic map were identified and combined into search strings using wildcards (*) and connectors (AND and OR). The wildcard (*) enables the search to detect multiple word endings (e.g., model* would detect model, models, modelling, etc.). Search terms were combined using the operator AND (both terms must be present somewhere in the search field) and OR (at least one of the terms must be present in the search field). This allows the search terms to be structured according to four thematic blocks: “Crop”; “Modelling”; “Plant disease”; and “Topic to exclude”.

The thematic block “Crop”, which consists of the common name of the crop (e.g., wheat or apple) or crop groups (e.g., citrus, berries, or nuts), was combined with the search strings of the other thematic blocks with the operator AND in the final searches: (crop (block) AND modelling (block) AND plant disease (block)) AND NOT topic to exclude (block)). The full search strings of the three thematic blocks (“Modelling”, “Plant disease”, and “Topic to exclude”) are presented in Table 1.

The full search was separately performed in the databases for each of the crops. The investigated crops were selected based on FAO data [24] about the production of all primary crops for all countries and regions in the world updated to 2019. For each crop and country, the area harvested (ha), the yield (kg), and the harvested production per unit of harvested area (hg/ha) were recorded. Based on these criteria, 103 crops or crop groups were selected.

The papers obtained from the first search in each of the selected databases were imported, and a separate file (library) was created for each crop. When the search was completed, all of the database libraries were incorporated into one new library, and duplicates were removed.

### 3.2. Paper Screening and Inclusion Criteria

The titles of all of the papers in the library were independently screened by each of us (authors) to remove those papers that failed to do the following:consider at least one of the selected crops;report about any type of model (e.g., empirical or mechanistic);concern any type of model purpose (e.g., scenario analysis, disease prediction, or crop protection);consider any step of model development (e.g., mathematical structure, evaluation, or practical implementation);focus on plant diseases, their causal agents (fungus, bacteria, virus, or phytoplasma), or their vectors.

If there was disagreement among the authors on the inclusion of a paper, the authors considered whether the inclusion/exclusion criteria required redefining.

### 3.3. Data Coding Strategy

Standardized descriptive metadata from all papers meeting the inclusion criteria were stored in a csv file, which formed the database for the systematic map. Data from each paper was coded as follows:Bibliographic information: Authors, title, year of publication, journal, publisher, reference type, language, number of citations, URL or DOI, affiliations of the corresponding author;Crop: Common name, code for crop groups. According to the Indicative Crop Classification (ICC) codes [25], crops were divided into the following crop systems: cereals, vegetables and melons, fruits (including nuts), oilseed crops, root and tuber crops, beverage and spice crops, leguminous crops, sugar crops, and other crops (Table 2);Location of study: Country. The affiliation of the corresponding author was used to define the location of the study. The countries were listed as indicated by the FAO based on the ISO 3166 international standard [26];Disease-causing organism: Scientific name of the causal agent, vector, or disease when specified in the title, and kingdom of the causal agent. Papers in which the disease-causing organism was not specified in the title were coded as “generic”;Study scope: Based on the title of the model, each model was assigned to the following categories according to its scope and purpose: (i) model for system representation and understanding; (ii) model for tactical disease management; (iii) model for strategic planning; and (iv) model for scenario analysis. The main characteristics and examples of papers for each category are listed in Table 3. These categories were described based on the terminology and contributions of Zadoks, Rabbinge, and Rossi [8,27,28].

## 4. Results

### 4.1. When Were Papers on Plant Disease Models Published?

The earliest paper we found was published in 1955, and only three and eighteen papers were published in the 1960’s and 1970’s, respectively (Figure 2). In the next decades, there was a substantial increase in the number of papers published on plant disease models, with almost 150 papers published in the 1990’s. The highest number of papers (*n* = 242) were published between 2010 and 2020 (Figure 2). In 2020 and 2021, 57 papers were published. The average number of papers/year in each decade increased from 2.25 in the 1970’s to 14.8 in the 1990’s, 21.8 in the 2000’s, 24.2 in the 2010’s, and 28.5 in the 2020’s. In total, 768 papers were included in the database for the systematic map.

### 4.2. In Which Countries Have Plant Disease Models Been Developed?

As indicated by the addresses of the corresponding authors, 59 countries have published papers concerning plant disease models (Figure 3). For North and South America, most studies were carried out in the USA (*n* = 280), followed by Brazil (*n* = 40) and Canada (*n* = 27). Most of the studies conducted in Europe originated from the UK (*n* = 60), France (*n* = 47), Germany (*n* = 34), Italy (*n* = 33), and Spain (*n* = 16). Among the Asian countries, most of the studies originated from India (*n* = 60), China (*n* = 38), and South Korea (*n* = 13). Australia and New Zealand contributed 32 and 16 papers, respectively. For Africa, studies have been conducted in Egypt (*n* = 1), Ethiopia (*n* = 1), Ivory Coast (*n* = 1), Kenya (*n* = 1), South-Africa (*n* = 2). 

### 4.3. Which Crops Were Targeted in Disease Modelling?

Papers on disease models have been published for a wide range of crops (Table 2). The main crop systems have been cereals and fruits (*n* = 214 and 210, respectively); although the number of papers was similar for cereal and fruits, the number of ha cultivated was much greater for cereals (36.82 billion) than fruits (1.72 billion) (Figure 4). In the case of oilseed crops, vegetables, and tubers, although their global cultivation areas range from 2.14 to 5.31 billion ha and exceed that of fruits, the number of papers on modelling was lower than for fruits, with *n* = 86 for oilseed crops, *n* = 85 for root and tubers crops, and *n* = 79 for vegetables and melons (Figure 4). The number of papers published per area cultivated was much lower for leguminous crops (*n* = 27), which are cultivated on > 1 billion ha, than for sugar (*n* = 23) and beverage/spice crops (*n* = 13), which are cultivated on only 0.23 and 0.11 billion ha, respectively.

Figure 5 indicates the crops that were represented by >15 papers in our database. These crops mainly belong to fruit crop systems, followed by cereal crop systems; only a few belonged to vegetable crop systems. Figure 5 also shows the relationship between the number of papers and the cultivated area (A) and global value in $ (B) of each crop. More papers were published for high than for low value crops, regardless of their cultivated area. Note that important crops, like maize and legumes, are not included in Figure 5, as those crops represented <15 papers in our database.

The crops represented by the largest number of papers were wheat (*n* = 143) and potatoes (*n* = 80), which differ more in the cultivated area (215 vs. 25 million ha, respectively) than in produce value (168 vs. 95 billion $, respectively). Grapes (*n* = 51) and apples (*n* = 44) were also highly represented, and have little cultivated area (6.9 and 4.7 million ha, respectively) yet high produce value (each with > 90 billion $). For soybean and rice, with 120 and 162 million of cultivated ha, respectively, and 310 and 100 billion $ of produce value, respectively, there were only 38 and 30 papers, respectively. More papers were published for rapeseed (*n* = 33) than for rice, although the area cultivated (16 million ha) and produce value (36 billion $) were lower. A similar number of papers were published for citrus fruits (*n* = 28), which are cultivated on only 8 million ha, although with a similar produce value of 39 billion $.

For nuts and berries, the number of papers published was similar (*n* = 20) and the produce value was not very different (36 vs. 28 billion $, respectively), however, the area cultivated was substantially greater for nuts than for berries (39 vs. 6 million ha, respectively). Few papers were also published concerning tomatoes (*n* = 19), onion (*n* = 18), sugar beet (*n* = 17), and barley (*n* = 17), which are grown on 5 million ha (except barley, which is grown on 51 million ha), and with values ranging from 92 billion $ for barley to 10.6 billion $ for sugar beet (except for tomatoes, with a value of 93 billion $).

### 4.4. Which Pathogen Kingdoms Were Considered?

The number of papers per pathogen kingdom are listed in Table 4. Fungi were the most studied kingdom, with more than 500 papers, followed by Chromista (*n* = 101). Forty-eight papers were published concerning viruses, and 17 papers for their vectors. A total of 44 papers were published for Bacteria, three of which focused on bacterial vectors. Animalia and Protista were considered in two and four papers, respectively. In total, 51 papers were classified as generic as the kingdom of the harmful organism (s) was not mentioned in the title, or as the models were not parametrized for specific pathosystems.

### 4.5. For Which Scope the Models Were Developed?

Figure 6 shows the main scopes of the papers concerning disease models (scopes are described in Table 3). The majority of the papers had models focused on system understanding (*n* = 680), i.e., models that considered the effect of environment and/or agronomic variables on pathogen/disease development, or that determined the validity of models under epidemiological or cropping conditions different from those under which the models had been developed. Forty papers concerned the use of disease models for tactical management (i.e., scheduling and timing of crop protection interventions) and 38 concerned strategic planning (i.e., evaluation of disease risk distribution); only nine papers concerned the use of disease models as simulators and for scenario analysis (Figure 6).

In the case of fruit crops, 171 papers focused on models for system understanding, 18 for tactical management, and 17 for strategic planning. For cereals, 193 papers focused on system understanding, nine on tactical management, and eight on strategic planning. For vegetables, oilseeds and tubers, >80 papers considered models for system understanding, and only two to five for tactical management or strategic planning. For legumes, 25 papers focused on models for understanding the pathosystem, and only one for strategic planning; no papers were found for tactical disease management. For beverage/species, a low number of papers focused on models for system understanding, tactical management, and strategic planning. For sugar crops, all 23 papers focused on system understanding (Figure 6).

## 5. Implications for Future Development of Plant Disease Models

Our systematic map included a large database of 768 research papers concerning “Disease modelling for agricultural crops worldwide”. The number of relevant papers published in each decade has steadily increased over time, and the number of papers published in 2020 and 2021 suggests that the development of plant disease models will continue to increase in the future. This trend likely reflects the development of disease models as a basis for decision making in crop protection via IPM and the need for more sustainable agricultural systems [4].

Although systematic maps follow rigorous, objective, and transparent processes to minimize bias [23], we cannot exclude the possibility that our systematic map may have some biases. The use of English as a language for searching and coding the papers could favor the papers from UK and USA (52 of the 768 papers in our database were written in other languages), and probably favor those pathosystems that are the main concern in those countries. The databases used for our literature search (WOS and Scopus) could affect the number of papers retrieved in the different decades, as these databases do not include some journals (like IOBC or EPPO Bulletins), in which models may have been published in the 1980’s and 1990’s. However, we decided to restrict the search to WOS and Scopus in order to provide a database of papers easily accessible to the research community.

The systematic map highlights that the number of papers on plant disease models is more affected by the economic value of the crop than by its cultivated area; this was the case, for example, for wheat, potatoes, grapes, and apples. Therefore, there is a research gap in disease modelling for crops that are cultivated over a large area worldwide, yet have a relatively low produce value, such as soybeans or nuts. The map also shows the underrepresentation in the development of disease models for maize and rice, which are extensively cultivated, have a high economic value, and are globally important foods for humans [48]. Underrepresentation of these crops in the modeling literature seems unrelated to the impact of diseases on their yield, i.e., both maize and rice yields and quality can be seriously reduced by disease. Oerke [49] and Savary et al. [50] mapped the effects of harmful organisms on major food crops and detected higher yield losses for maize and rice (30% and 22.5%) than for wheat or potatoes (21.5 and 17.2%). However, these authors found important differences in the crop health among world regions, with lower crop losses in regions that generate food surpluses, and higher losses in food-insecure regions. Our work highlights a not uniform distribution of papers on disease models at a global scale, with countries in North America or Europe producing more disease modeling papers than those in Asia or Africa. Although this could reflect biases, the relatively low attention to the development of rice disease models could also be related to the low number of researchers working on modelling in southeast Asia where rice cultivation is prevalent. This, however, cannot be true for maize, which is an important crop in North America and Europe [48].

Another interesting result of the current research is the low number of models for diseases of legumes. Legumes have a key role in addressing future global food security and environmental challenges, as well as in contributing to human health [51]. Lack of models for efficient disease management in legumes could contribute to the lower rate of increase in both the production and economic value of legumes than of cereals and other crops [52].

As previously observed by de Wolf and Isaard [15], trends in the development of plant disease models are strongly influenced by a variety of scientific and social factors, as well as by the emergence (or re-emergence) of important diseases that greatly affect crop yield and quality. Considering the scope of plant disease models, papers concerned with system understanding are much more numerous than papers concerned with scenario analysis, tactical disease management, or strategic planning.

The numbers of papers addressing disease management (in terms of tactical management and strategic planning) is consistent with the observation of past reviews of plant disease models, which indicates that far more models are developed than are applied for practical disease management [15,16,20,53,54]. Models for system understanding are built to describe disease cycles, drivers of epidemic development, and host-pathogen relationships, and to enable the simulation of important steps in pathogen development (e.g., sporulation, infection, and disease onset) and plant disease epidemics; the information generated from such models is essential for disease management [8,15]. Modelling for tactical management and strategic planning may capitalize on models for system understanding in order to develop risk algorithms, intervention thresholds, decision rules, or decision support systems that help growers and advisors in decision making for crop protection [4,55]. Efforts are therefore needed to extend models for system understanding into models that support decision making. For instance, a mechanistic model simulating primary infections of downy mildew on grapevines [56] has been validated over different years, and grape-growing areas, for both its ability to correctly predict the disease [57] and to schedule fungicide applications [58] before its integration into vite.net^®^, which is a decision support system (DSS) [59].

A limitation of the current study is that in the development of the systematic map, we classified papers according to model, scope, and purpose based only on paper titles; it is therefore possible that some papers included additional scopes or purposes that were not indicated in their titles. For example, Rossi et al. [60,61] deal with the prediction of *Venturia inaequalis* primary infections on apples; although we classified the scope of both papers as system understanding (as they consider the effect of environment on disease development), Rossi et al. [61] also provided a risk index that can be used to schedule fungicide sprays. Therefore, their model considered tactical disease management in addition to system understanding.

In summary, our systematic map has identified a variety of areas for which there are knowledge gaps and has suggested questions that warrant further research. In particular, our systematic map has provided data that support two commonly made inferences: first, additional attention should be paid to the application of models for practical disease management; and second, increased efforts should be directed at development of disease models for essential crops (e.g., rice and maize). Moreover, our systematic map will help researchers identify cropping systems and locations that are sufficiently represented in the literature database to enable systematic review.

## Figures and Tables

**Figure 1 plants-11-00724-f001:**
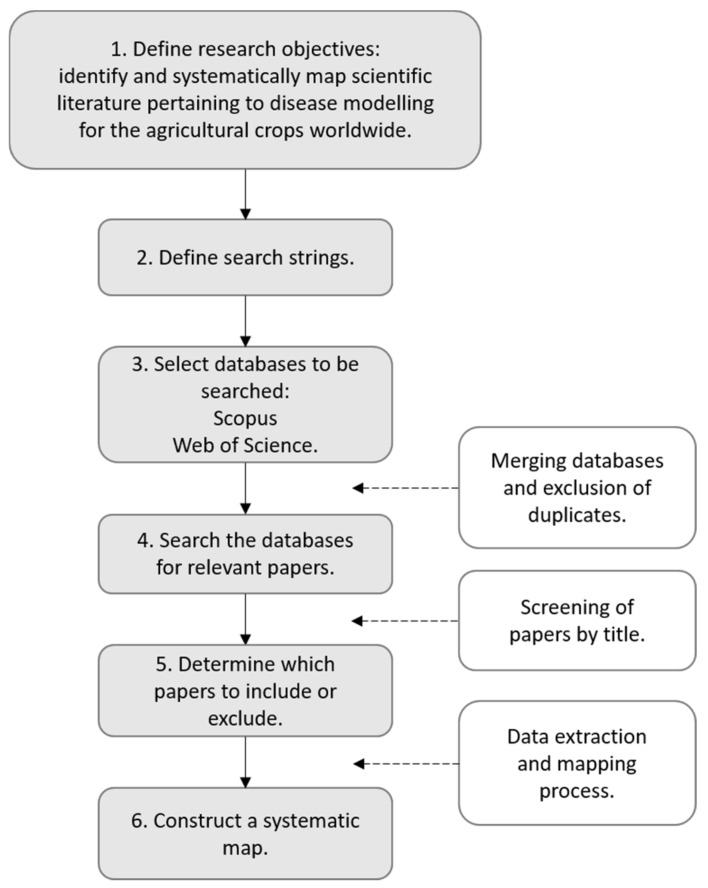
Steps for the development of the systematic map.

**Figure 2 plants-11-00724-f002:**
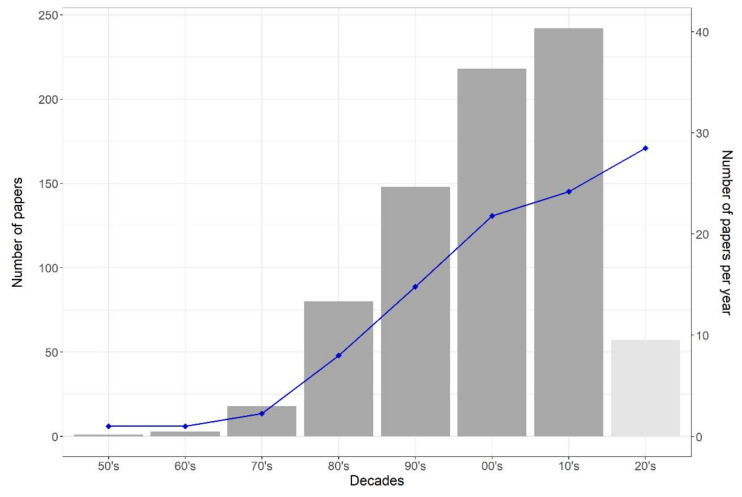
Number of papers per decade. The bars indicate the total number of papers retrieved in each decade, from the 1950’s to the 2020’s. The blue points indicate the average number of papers per year retrieved in each decade. The papers corresponding to the 2020’s consider only the years 2020 and 2021 (light grey).

**Figure 3 plants-11-00724-f003:**
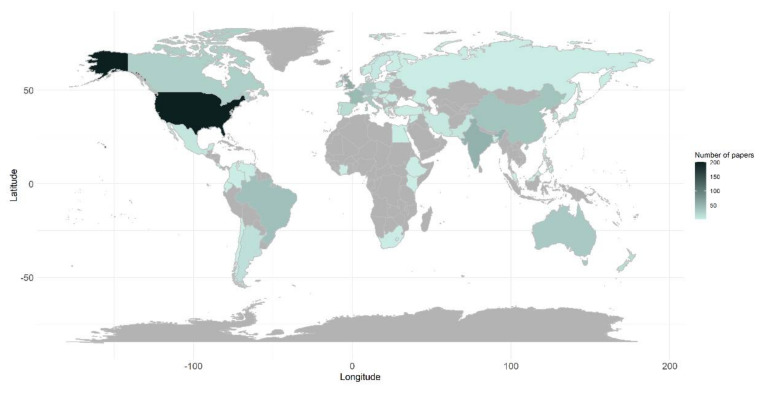
Country of origin of the studies. The total number of papers per country is shown as a color gradient, from light green-blue (lower number of papers) to dark green-blue (higher number of papers). Countries for which no corresponding authors have been found in the literature are indicated by grey.

**Figure 4 plants-11-00724-f004:**
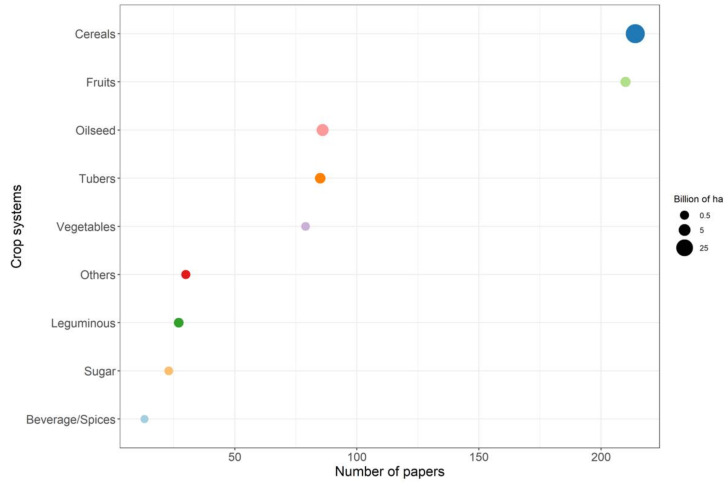
Number of papers grouped by crop system. The dot size indicates the global cultivated area for each crop system.

**Figure 5 plants-11-00724-f005:**
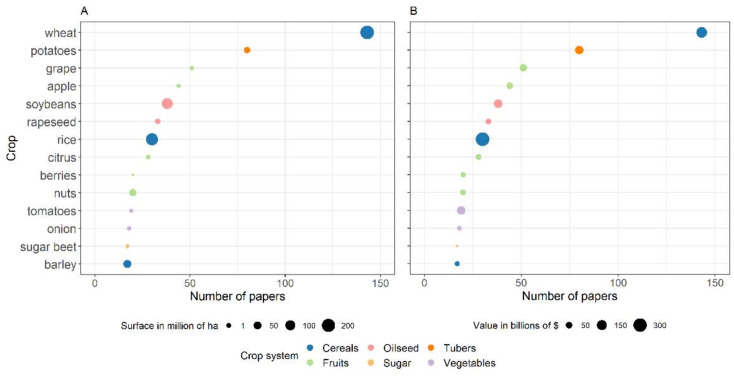
Crops for which > 15 papers concerning disease models have been published. The dot size indicates the global cultivated area (**A**), and the produce value in $ (**B**); the dot color defines the crop system.

**Figure 6 plants-11-00724-f006:**
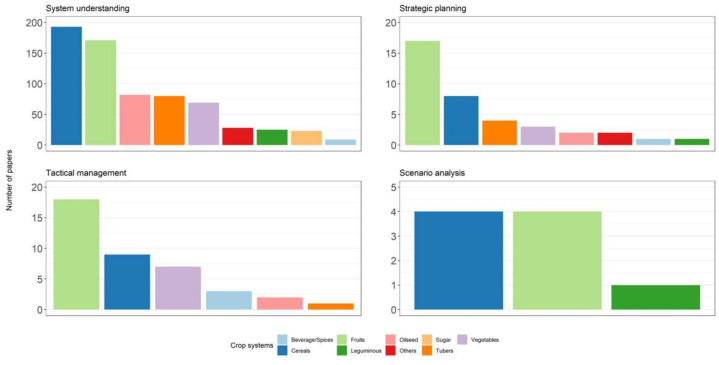
Number of papers for each scope of disease models and for each crop group. Scopes are described in Table 3.

**Table 1 plants-11-00724-t001:** Search strings for three thematic blocks.

Thematic Block	Search Strings
Modeling	model* OR simulat* OR predict* OR forecast* OR prognos*
Plant disease	disease* OR pathog* OR epidem* OR infect*
Topic to exclude	molec* OR gen* OR “image recognition” OR weed OR locus OR Arabidopsis OR Brachypodium OR cell OR human OR celiac OR coeliac OR cancer OR allergy OR hyper* OR rat OR mouse

**Table 2 plants-11-00724-t002:** Crops belonging to crop systems according to the Indicative Crop Classification (ICC) codes [25], and the number of selected papers for each crop (in brackets).

Crop System	Crops (Number of Selected Papers)
1. Cereals	Barley (17), Maize (15), Millet (1), Oats (2), Rice (30), Rye (2), Sorghum (4), Wheat (143)
2. Vegetables and melon	Artichokes (1), Asparagus (2), Brassicas (12), Carrots (9), Cucumbers (7), Eggplants (1), Garlic (1), Leeks (1), Lettuce (5), Melon (1), Onion (18), Quinoa (1), Tomatoes (19), Watermelon (1)
3. Fruits and nuts	Apple (44), Banana (8), Cherries (2), Citrus (28), Grape (51), Kiwi (1), Mango (7), Nectarines (6), Nuts (20), Papaya (1), Pears (13), Pineapple (1), Pistachios (2), Plantain (4), Plums (2), Strawberries (20)
4. Oilseed crops	Coconut (2), Mustard (4), Oil palm (2), Olives (3), Rapeseed (33), Safflower (1), Soybeans (38), Sunflower (3)
5. Root and tuber crops	Cassava (5), Potatoes (80)
6. Beverage and spice crops	Chillies (5), Cocoa (1), Coffee (7)
7. Leguminous crops	Beans (15), Lentils (1), Peas (11)
8. Sugar crops	Sugar beet (17), Sugar cane (6)
9. Other crops	Cotton (11), Hops (9), Persimmon (1), Poppy (1), Rubber (2), Tobacco (6)

**Table 3 plants-11-00724-t003:** Assignment of models according to scope and purpose, and references of some examples of works for each scope.

Scope Category	Examples of Scope	Examples of Papers
1. System representation and understanding	1.1 Effect of environmental or agronomical variables on disease development 1.2 Simulation of epidemic development in time and/or space 1.3 Simulation of yield losses due to disease development 1.4 Evaluation or validation of previously developed models	[29,30,31,32,33,34,35]
2. Tactical disease management	2.1 Schedule of crop protection interventions 2.2 Best timing and frequency of disease control measures	[36,37,38,39,40]
3. Strategic planning	3.1 Evaluation of disease risk distribution (spatial, climatic, or geographic)	[41,42,43,44]
4. Scenario analysis	4.1 Simulation, interpretation, and evaluation of crop protection scenarios	[45,46,47]

**Table 4 plants-11-00724-t004:** Number of papers recorded for each kingdom.

Kingdom	No. of Papers
Fungi	501
Chromista	101
Generic ^1^	51
Bacteria (vector)	41 (3)
Virus (vector)	48 (17)
Protista	4
Animalia	2

^1^ Generic indicates those papers for which it was impossible to define the kingdom of the harmful organism (s) based on the title, or for papers with models that were not parametrized for specific pathosystems.

## Data Availability

The data that support the findings of this study are available on request from the corresponding author, V.R.

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
