# Peer review of "A Systematic Map of the Research on Disease Modelling for Agricultural Crops Worldwide"

_plants, 2022, doi:10.3390/plants11060724_

Round 1

Reviewer 1 Report

General comments

This manuscript presents an overview of the literature pertaining to disease modelling for agricultural crops worldwide. In total, 768 papers were retrieved reporting disease models for 108 crops. The main limitation of this manuscript is that the search was based only on paper titles. This can lead to not finding some published papers.  It would have been advisable to include in the search the keywords or summary and, also, to use some other topics in the search strings related to warning or decision support systems. Moreover, it is possible that some papers included additional scopes or purposes that were not indicated in their titles. The authors already acknowledge these limitations (section 5). However, the results obtained are very interesting, and the manuscript provides an excellent overview of disease models by crops, pathogens, countries and scopes.

Specific comments

Figure 1. Check the scheme, there are cut parts.

Figure 3. Revise legend colours. Dark green-blue seems black.  It would be better if the grey (indicating countries for which no corresponding authors have been found in the literature)  was white, or a colour very different from black (dark blue).

Line 200: The word ‘number’ is repeated.

Author Response

Dear Reviewer,

thank you very much for your revision.

We adressed your comments. Regarding Figure 3, we changed the colour for countries with any paper, using light grey. For the darker colour of the legend, we kept the definition "dark green-blue" because this is the name in the R code description of the palette.

Regards,

the authors

Reviewer 2 Report

Dear Authors,

Your manuscript quantifies our impressions about plant-pathological modelling. The data are interesting and motivating. I don't suggest any major changes, but please check over the reference list, because many bibliographic data are incomplete (e.g. ref. 41-61).

Author Response

Dear Reviewer,

thank you very much for your kind reply.

We checked carefully the references.

Regards,

the authors